# Preoperative nutritional evaluation of patients with hepatic alveolar echinococcosis

Xie Liang[1,2], Wang Shu[3], Zhou Linyong[4], Li Jianshui[1,2], Gu Junqing[3], Dawa Enzhu[4], Xu Mingqing[5]*

1 Department of Hepatobiliary Surgery (2), the Affiliated Hospital of North Sichuan Medical College, Nanchong, China, 2 Northeast Sichuan Acute Pancreatic Research Center, North Sichuan Medical College, Nanchong, China, 3 Department of Urology Surgery, the Affiliated Hospital of North Sichuan Medical College, Nanchong, China, 4 Surgery Department of the People's Hospital of Ganzi County, Ganzi Tibetan Autonomous Prefecture, China, 5 Department of hepatic surgery, West China Hospital of Sichuan University, Chengdu,China

* xumingqing@scu.edu.cn

## Abstract

### Objective

This study is aimed at determining the preoperative nutritional status of patients with hepatic alveolar echinococcosis (HAE), and subsequently establish a concise and reasonable nutritional evaluation indicator. The established evaluation method could be used for clinical preoperative risk assessment and prediction of post-operation recovery.

### Methods

The basic patient information on height, body weight, BMI and hepatic encephalopathy of 93 HAE patients were examined. Subsequently, abdominal ultrasonography, blood coagulation and liver function tests were done on the patients. Liver function was assessed using the Child-Pugh improved grading method while nutritional status was evaluated using the European Nutrition Risk Screening 2002 (NRS 2002) method. Additional parameters including hospitalization time, the hemoglobin (HGB) level on the 3rd day after the operation, and the number of postoperative complications of HAE patients were also recorded.

### Results

The NRS 2002 score was negatively correlated with body weight, body mass index (BMI) and albumin (ALB) (P<0.01), and positively correlated with the transverse and longitudinal diameters of the lesions (P<0.01). A worse grading of liver function was associated with a low ALB and a high NRS 2002 score (P<0.01). Results of the NRS 2002 score indicate that the hospitalization time of the normal nutrition group was significantly shorter than that of the malnourished group (P < 0.05). The HGB level of the control group on the 3rd day after the operation was significantly higher than that of the malnourished group (P < 0.05), and the number of postoperative complications was lower than that of malnutrition group (P < 0.05).

**Data Availability Statement:** All relevant data are in the paper and Supporting Information files.

**Funding:** This research is supported by the project of Sichuan Province health and Family Planning Commission (17PJ106)

**Competing interests:** The authors have declared that no competing interests exist.

## Conclusion

Malnutrition is common in HAE patients. The nutritional status of HAE patients is related to many clinical factors, such as Child-Pugh classification of liver function, size of the lesion, and ALB among others. Although both BMI and ALB can be used as primary screening indicators for malnutrition in HAE patients, NRS 2002 is more reliable and prudent in judging malnutrition in HAE patients. Therefore, BMI and ALB are more suitable for preoperative risk assessment and prediction of postoperative recovery.

## Introduction

Hepatic echinococcosis (HE) is an endemic helminthic disease categorized into hepatic cystic echinococcosis (HCE) and hepatic alveolar echinococcosis (HAE). Among the two forms of HE, HAE is the most life-threatening. Hepatic alveolar echinococcosis (HAE) is caused by infection with the *Echinococcus multilocular* helminth [1, 2], and accounts for 3% of the total number of human echinococcosis [3]. The disease is characterized by a slow but concealed onset, and invasive growth, similar to hepatocellular carcinoma. Notably, HAE is commonly known as "worm cancer" and "parasitic liver cancer" [4]. In China, HAE is more prevalent in the Tibet Autonomous Region, Qinghai Province, and the Ganzi Tibetan Autonomous Prefecture. The disease is a chronic consumptive disease. Damage to the liver impairs the synthesis and metabolism of nutrients such as albumin. A decrease in blood albumin levels results in a reduction in body weight, a poor general condition of patients and the inability to tolerate surgical treatment [5].

Nutrition refers to the process by which the human body ingests and metabolizes food through digestion, absorption, and metabolism to maintain life activities. Nutrition forms the basis of sustaining normal physiological functions of the human body. Proper nutrition is crucial to tissue repair, and the provision of active immunity and resistance to diseases. Malnutrition refers to the insufficient intake or absorption of nutrients by the body caused by hunger, illness, aging and other factors. These factors lead to a decrease in body composition (fat-free cell population), changes in the somatic cell population, and a reduction in physiological function, which cause adverse clinical outcomes of patients [6]. The incidence of malnutrition in surgical patients ranges from 20% to 60%. Malnutrition compromises the body's immune resistance to stressful events such as surgery and infection. Malnutrition also damages the function of body organs and tissues, increases the incidence of complications and mortality after the operation, increases medical expenses, prolongs hospitalization time, and affects the clinical outcomes of patients [7, 8].

Hepatic alveolar echinococcosis (HAE) operation is characterized by long operation time, high operation difficulty, many postoperative complications and slow postoperative recovery. As a result, the nutritional status of HAE patients before operation significantly affects the success or failure, and rapid recovery post-operation. Therefore, effective screening and diagnosis of malnutrition in HAE patients before the operation and provision of active interventions are paramount to patient recovery and significantly improves prognosis [9]. These findings are also in line with the concept of treatment and rehabilitation of Enhanced Recovery After Surgery (ERAS), which emphasizes perioperative nutritional support [10]. At present, there is no convincing and compelling evaluation system of the preoperative dietary status of HAE patients. This study explored the preoperative nutritional status of HAE patients to establish concise and practical nutritional evaluation indicators for preoperative risk assessment. The

study also evaluated the relationship between nutritional indicators, general conditions, clinical indicators, and postoperative recovery indicators of HAE patients.

## Objects and methods

### Objects

A total of 93 patients diagnosed with HAE during a hepatic echinococcosis screening project in Ganzi People's Hospital in April 2019 were included in this study. The study participants included 42 males and 51 females. The inclusion criteria included Tibetan patients diagnosed with HAE, of sound mind and could respond to questions appropriately, and who were available for the entire study period. The exclusion criteria included patients with malnutrition caused by other previous diseases and patients with HCE or other liver lesions such as hepatic hemangioma and hepatocellular carcinoma. Also excluded were patients with infectious diseases (respiratory tract infection, pulmonary infection, etc.) or other chronic consumptive diseases, and patients with other organ echinococcosis.

### Methods

This study has been approved by the Medical Ethics Committee of the Affiliated Hospital of North Sichuan Medical College. All patients have been informed suitably, and we've asked for their verbal consent. We've obtained consent from parents or guardians when our study included minors under age 18.

**Inspection indicators.** The basic information of height, weight, occupation, nationality and information on the existence of hepatic encephalopathy were recorded in all patients. The location, number, transverse diameter, longitudinal diameter and the ascites status of hepatic alveolar echinococcosis (HAE) masses were assessed by abdominal ultrasonography. Blood routine, coagulation routine and liver function tests were performed to determine the patients' hemoglobin (HGB), aspartate aminotransferase (AST), alanine aminotransferase (ALT), total bilirubin (TBIL), albumin (ALB) and prothrombin (PT). The hospitalization time, HGB value on the third day after the operation and the number of postoperative complications (including wound liquefaction and infection, lung infection, abdominal and liver wound infection, urinary tract infection, postoperative inflammatory intestinal obstruction, bile leakage and acute liver function injury) were recorded.

**Classification of the standard of liver function.** The Child-Pugh improved grading method was used for grading liver function [11]. The grading of liver function was calculated by measuring TBIL, ALB, PT, ascites and hepatic encephalopathy. The Child-Pugh grade A patients, 5–6 points, had a better liver function, grade B, 7-9points, had a moderate liver function, while grade C, 10–15 points, had severe damage on liver function.

**Evaluation method of nutritional status.** A body mass index (BMI) $\geqq 24.0(\text{Kg/m}^2)$ was defined as overweight, while BMI $< 18.5$ $(\text{Kg/m}^2)$ was defined as malnutrition. The nutritional status of patients was assessed by the European Nutrition Risk Screening 2002 (NRS 2002) [12]. Nutritional status parameters included recent weight changes, BMI, the severity of disease and feeding status. Patients with a total score $\geqq 3$ were assessed as having malnutrition risk (need to formulate reasonable clinical nutrition support plan) while an overall NRS score $<3$ was interpreted as no nutrition risk hence did not require clinical nutrition support, but concurrent screens were needed [13].

**Statistical methods.** All data were input and analyzed by statistical software SPSS17.0 (SPSS Inc., Chicago, IL, USA). Continuous data were presented as Mean±standard deviation. Correlations between nutritional indicators and patients' general condition, and ultrasonography results and hematological indicators were determined using Pearson correlation analysis.

**Table 1. General conditions of patients.**

| Indicator | Group | N | Percentage |
|---|---|---|---|
| **sex** | male | 42 | 45.16% |
| | female | 51 | 54.84% |
| **BMI** | Overweight (BMI < 24) | 41 | 44.09% |
| | Normal (18.5 < BMI < 24) | 46 | 49.46% |
| | Malnutrition (BMI < 18.5) | 6 | 6.45% |
| **NRS 2002 score** | ≥3 | 34 | 36.56% |
| | <3 | 59 | 63.44% |
| **Classification of liver function** | Grade A group | 58 | 62.37% |
| | Grade B group | 29 | 31.18% |
| | Grade C group | 6 | 6.45% |
| **ALB** | Normal (ALB≥35g/L) | 81 | 87.10% |
| | Malnutrition ALB<35g/L) | 12 | 12.90% |
| **location of liver hydatid lesions** | Right lobe | 65 | 69.90% |
| | Left lobe | 11 | 11.83% |
| | Both left and right lobes | 17 | 18.28% |
| **number of liver hydatid lesions** | 1 | 73 | 78.50% |
| | 2 | 11 | 11.83% |
| | ≥3 | 9 | 9.68% |

The LSD test was used for multiple comparisons of liver function classification and nutritional indicators. Student's T-test was used to compare the postoperative recovery indicators between groups.

## Results

### General conditions of study objects

A total of 93 patients aged 12–81, and of Tibetan origin were included in this study. The patients had an average age of 44.66±14.40 years; 42 males (45.16%) and 51 females (54.84%) height range of 140–183 cm, and an average height of 162.45±9.223cm. The weight of included patients ranged between 34 and 90 kg, with an average weight of 63.33±11.690kg. Data on the BMI, NRS 2002 score, liver function classification, serum ALB value, location, and the number of liver hydatid lesions are as shown in Table 1. The transverse and longitudinal diameters of liver hydatid lesions, serum ALT, AST, ALB, TBIL and blood HGB values are presented in Table 2.

**Table 2. Ultrasonography and hematological indicators of patients.**

| Indicator | N | min | max | Mean | Std. Deviation |
|---|---|---|---|---|---|
| **transverse diameters (cm)** | 93 | 1.2 | 16.4 | 5.957 | 3.2631 |
| **longitudinal diameters (cm)** | 93 | 1.1 | 17.2 | 4.894 | 2.8891 |
| **ALT (u/L)** | 93 | 3.0 | 144.0 | 39.989 | 30.0077 |
| **AST (u/L)** | 93 | 7.0 | 131.0 | 23.931 | 18.2513 |
| **ALB (g/L)** | 93 | 28.5 | 55.8 | 42.569 | 6.5359 |
| **TBIL (umol/L)** | 93 | 5.2 | 65.0 | 11.968 | 8.0408 |
| **HGB (g/L)** | 93 | 110 | 192 | 151.59 | 21.257 |

**Table 3. Pearson correlation analysis between nutritional indicators and general conditions and morphological indicators of lesions.**

| | | NRS2002 score | BMI | ALB | body weight | height | age | location of liver hydatid lesions | number of liver hydatid lesions | transverse diameters | longitudinal diameters |
|---|---|---|---|---|---|---|---|---|---|---|---|
| **NRS 2002 score** | **Pearson correlation** | 1 | -.338** | -.481** | -.352** | -.123 | .030 | .161 | .129 | .348** | .338** |
| | **Sig. (2-tailed)** | | .001 | < .001 | .001 | .241 | .776 | .124 | .220 | .001 | .001 |
| | **N** | 93 | 93 | 93 | 93 | 93 | 93 | 93 | 93 | 93 | 93 |
| **BMI** | **Pearson correlation** | -.338** | 1 | .341** | .789** | .002 | .359** | .142 | .117 | -.088 | -.100 |
| | **Sig. (2-tailed)** | .001 | | .001 | < .001 | .985 | < .001 | .174 | .265 | .402 | .342 |
| | **N** | 93 | 93 | 93 | 93 | 93 | 93 | 93 | 93 | 93 | 93 |
| **ALB** | **Pearson correlation** | -.481** | .341** | 1 | .372** | .136 | .151 | -.078 | -.070 | -.360** | -.372** |
| | **Sig. (2-tailed)** | < .001 | .001 | | < .001 | .193 | .149 | .459 | .504 | < .001 | < .001 |
| | **N** | 93 | 93 | 93 | 93 | 93 | 93 | 93 | 93 | 93 | 93 |
| **body weight** | **Pearson correlation** | -.352** | .789** | .372** | 1 | .602** | .221* | .047 | .055 | -.055 | -.062 |
| | **Sig. (2-tailed)** | .001 | < .001 | < .001 | | < .001 | .034 | .654 | .597 | .599 | .558 |
| | **N** | 93 | 93 | 93 | 93 | 93 | 93 | 93 | 93 | 93 | 93 |

**. Correlation is significant at the 0.01 level (2-tailed).

*. Correlation is significant at the 0.05 level (2-tailed).

## Analysis of the correlation between nutritional indicators and general conditions, ultrasonography and hematological indicators of patients

Results on the *Pearson* correlation analysis between nutritional indicators general conditions and morphological indicators of the lesion are shown in Table 3. It was found that the NRS2002 score was negatively correlated with body weight, BMI and ALB (P < 0.01), and positively correlated with the transverse and longitudinal diameter of the lesion (P < 0.01). The body mass index (BMI) was positively correlated with serum ALB, body weight and age (P < 0.01). Serum ALB was positively associated with body weight and BMI (P < 0.01), and negatively correlated with transverse and longitudinal diameters of lesions (P < 0.01).

The *Pearson* correlation analysis of nutritional indicators and hematological test indicators are shown in Table 4. It was found that there was no significant correlation between NRS 2002 score, BMI, serum ALB and AST, ALT, TBIL, HGB value in the blood (P > 0.05), and there was a significant positive correlation between body weight and blood ALT value (P < 0.01).

The results of LSD multiple comparisons between liver function classification and nutritional indicators are presented in Table 5. It was found that there were significant differences in NRS2002 score, body weight, BMI and serum ALB between Grades A and B patients (P < 0.01), and also between Grades A and C patients. The NRS2002 score and serum ALB of Grade B patients significantly differed from the scores of Grade C patients (P < 0.05).

## Comparative analysis of postoperative recovery indicators between normal nutrition group and malnutrition group

A total of 74 patients (79.57%) completed the operation, while the other 19 patients (20.43%) were not admitted to the hospital because of various complexities surrounding the operation (7.45%, n = 7). The complexities included family factors (5.32%, n = 5), religious factors

**Table 4. Pearson correlation analysis between nutritional indicators and hematological indicators.**

| | | NRS 2002 score | BMI | ALB | body weight | ALT | AST | TBIL | HGB |
|---|---|---|---|---|---|---|---|---|---|
| **NRS 2002 score** | **Pearson Correlation** | 1 | -.338** | -.481** | -.352** | .073 | .196 | .155 | -.156 |
| | **Sig. (2-tailed)** | | .001 | < .001 | .001 | .485 | .059 | .137 | .136 |
| | **N** | 93 | 93 | 93 | 93 | 93 | 93 | 93 | 93 |
| **BMI** | **Pearson correlation** | -.338** | 1 | .341** | .789** | .188 | -.024 | -.121 | -.120 |
| | **Sig. (2-tailed)** | .001 | | .001 | < .001 | .071 | .820 | .250 | .253 |
| | **N** | 93 | 93 | 93 | 93 | 93 | 93 | 93 | 93 |
| **ALB** | **Pearson correlation** | -.481** | .341** | 1 | .372** | .101 | -.124 | -.145 | -.022 |
| | **Sig. (2-tailed)** | < .001 | .001 | | < .001 | .333 | .237 | .166 | .831 |
| | **N** | 93 | 93 | 93 | 93 | 93 | 93 | 93 | 93 |
| **body weight** | **Pearson correlation** | -.352** | .789** | .372** | 1 | .355** | .051 | -.042 | -.013 |
| | **Sig. (2-tailed)** | .001 | < .001 | < .001 | | < .001 | .627 | .691 | .898 |
| | **N** | 93 | 93 | 93 | 93 | 93 | 93 | 93 | 93 |

**. Correlation is significant at the 0.01 level (2-tailed).

*. Correlation is significant at the 0.05 level (2-tailed).

(4.26%, n = 4), economic factors (2.13%, n = 2), and remote medical treatment (1.06%, n = 1). Details on the grouping characteristics and the comparative analysis of postoperative recovery indicators between the normal nutrition group and the malnutrition group are presented in Tables 1 and 6, respectively. As shown in Table 6, the NRS 2002 nutrition scores indicate that there were no statistical differences in pre-operation HGB levels between the two groups (P>0.05). Compared to the normal nutrition group, hospitalization time and HGB levels on the third day after operation were significantly shorter (P < 0.05), and significantly higher (P < 0.05), respectively than in the malnutrition group. Also, the number of postoperative complications was lower than that of the malnutrition group (P < 0.05). However, no significant differences in BMI and ALB were noted between the two groups (P > 0.05).

## Discussion

### Selection of nutritional indicators

Four nutritional indicators, including body weight, body mass index (BMI), albumin (ALB) and NRS2002 score values, were selected based on the ease of clinical access. Although body weight is one of the most intuitive nutritional indicators, it can only reflect one aspect of human characteristics. Assessment of the body mass index (BMI), which also considers the height factor, has become the most widely used nutritional evaluation method. According to the 2015 European Society for Parenteral and Enteral Nutrition (ESPEN) proposal, both weight and BMI should be used as diagnostic indicators of malnutrition [14]. However, long-term clinical practice has proved that the nutritional status of patients can conveniently be assessed by BMI alone, although significant differences in the reliability of BMI for different populations has been reported [15]. Individual differences and dietary habits affect the relationship between BMI and body function; hence, the composition of the human body is not constant. For example, the patients included in this study are Tibetans who dwell in pastoral areas. Their physical fitness, dietary habits, lifestyle, and exercise intensity are quite different from those of the Han nationality. Therefore, the BMI of Tibetans is generally higher than their actual nutrition. Among such individuals, some deviations exist in the status, and it is difficult to accurately reflect recent changes in body weight and preoperative nutritional status.

**Table 5. Multiple LSD comparisons of liver function classification and nutritional indicators.**

| dependent variable | (I) liver function classification | (J) liver function classification | Mean Difference (I-J) | Std. Error | Sig. | 95% Confidence Interval | |
|---|---|---|---|---|---|---|---|
| | | | | | | Lower Bound | Upper Bound |
| **NRS 2002 score** | A | B | -1.828* | .245 | < .001 | -2.31 | -1.34 |
| | | C | -2.856* | .463 | < .001 | -3.78 | -1.94 |
| | B | A | 1.828* | .245 | < .001 | 1.34 | 2.31 |
| | | C | -1.029* | .484 | .036 | -1.99 | -.07 |
| | C | A | 2.856* | .463 | < .001 | 1.94 | 3.78 |
| | | B | 1.029* | .484 | .036 | .07 | 1.99 |
| **BMI** | A | B | 2.7931* | .7436 | < .001 | 1.316 | 4.270 |
| | | C | 5.1161* | 1.4022 | < .001 | 2.330 | 7.902 |
| | B | A | -2.7931* | .7436 | < .001 | -4.270 | -1.316 |
| | | C | 2.3230 | 1.4665 | .117 | -.590 | 5.236 |
| | C | A | -5.1161* | 1.4022 | < .001 | -7.902 | -2.330 |
| | | B | -2.3230 | 1.4665 | .117 | -5.236 | .590 |
| **body weight** | A | B | 9.621* | 2.404 | < .001 | 4.85 | 14.40 |
| | | C | 14.609* | 4.532 | .002 | 5.60 | 23.61 |
| | B | A | -9.621* | 2.404 | < .001 | -14.40 | -4.85 |
| | | C | 4.989 | 4.740 | .295 | -4.43 | 14.41 |
| | C | A | -14.609* | 4.532 | .002 | -23.61 | -5.60 |
| | | B | -4.989 | 4.740 | .295 | -14.41 | 4.43 |
| **ALB** | A | B | 8.9586* | .9574 | < .001 | 7.057 | 10.861 |
| | | C | 14.8006* | 1.8053 | < .001 | 11.214 | 18.387 |
| | B | A | -8.9586* | .9574 | < .001 | -10.861 | -7.057 |
| | | C | 5.8420* | 1.8881 | .003 | 2.091 | 9.593 |
| | C | A | -14.8006* | 1.8053 | < .001 | -18.387 | -11.214 |
| | | B | -5.8420* | 1.8881 | .003 | -9.593 | -2.091 |

*. The mean difference is significant at the 0.05 level.

Subsequently, this study incorporated the NRS 2002 score, which is grounded on many evidence-based medical reports and is closely related to clinical prognosis. The score is suitable for adult or elderly community patients and inpatients and has high sensitivity, better

**Table 6. A comparative analysis of postoperative recovery indicators between normal nutrition group and malnutrition group.**

| postoperative recovery indicators | NRS 2002 | | t | P | BMI | | t | P | ALB | | t | P |
|---|---|---|---|---|---|---|---|---|---|---|---|---|
| | Normal nutrition group (63.5%, n = 47) | Malnutrition group (36.5%, n = 27) | | | Normal nutrition group (93.2%, n = 69) | Malnutrition group (6.8%, n = 5) | | | Normal nutrition group (87.8%, n = 65) | Malnutrition group (12.2%, n = 9) | | |
| **hospitalization time (day)** | 10.4681 | 11.9259 | 4.030 | < .001* | 10.9855 | 11.2000 | -.279 | .781 | 10.9077 | 11.6667 | -1.302 | .197 |
| **HGB on the 3rd day after operation (g/L)** | 135.2340 | 120.9630 | -3.127 | .003* | 130.8696 | 118.4000 | 1.354 | .180 | 131.6308 | 118.4444 | 1.886 | .063 |
| **Postoperative complications (person time)** | .2128 | .4444 | 2.002 | .049* | .3043 | .2000 | .458 | .648 | .2923 | .3333 | -.234 | .815 |

*. The mean difference is significant at the 0.05 level.

specificity and a lower rate of false positives as compared to BMI. The NRS 2002 sore is a simple and highly operational screening method [16–18]. In this study, the NRS 2002 survey of hepatic alveolar echinococcosis (HAE) of pre-operation patients was completed when the patients were admitted for history-taking and physical examination. The assessment for each patient only took 5–10 minutes. As such, we identified the malnourished patients for the first time, without incurring any additional costs and without incorporating any invasive operation.

The liver is the main body organ involved in the synthesis of albumin (ALB), and HAE can affect protein synthesis in the liver. As a result, serum ALB levels reflect the severity of HAE. An ALB < 35g/L indicates hypoproteinemia and induces malnutrition. Also, persistent hypoproteinemia is an essential objective indicator of malnutrition [19]. Since ALB has a half-life of about 20 days, ALB levels can be conveniently used as a measure of chronic malnutrition.

## Relationship between nutritional status and postoperative recovery indicators of HAE patients

According to the results of the NRS 2002, BMI and ALB, HAE patients were divided into two groups: normal nutrition and malnutrition groups. Statistical differences in the postoperative recovery indicators between the groups were only observed in the NRS 2002 scores. The hospitalization time of the malnutrition group was longer, the hemoglobin level (HGB) was lower on the 3rd day after the operation, and the number of postoperative complications was higher. It can be seen that NRS 2002 is a reliable method to predict the postoperative recovery. According to the BMI or ALB results, the hospitalization time of the malnutrition group was slightly longer than that of normal nutrition group. Also, the HGB of the malnutrition group on the 3rd day after the operation was marginally lower than that of the normal nutrition group. However, the differences in both hospitalization time and hemoglobin levels between the two groups were not statistically significant. The lack of substantial differences in the two parameters could be caused by the small sample size used in our study, and the fact that BMI and ALB are less sensitive in the prediction of postoperative recovery.

Based on NRS 2002, malnourished HAE patients can be identified before operation by an NRS 2002 score ≥3. For such patients, substantial nutritional support and treatment should be provided before the surgery, and they suggested that the operation time should be postponed until the NRS 2002 scores of the patients improve. As such, the patient can better tolerate surgery and anesthesia, and the disease prognosis is improved. Also, the incidence of surgical complications and mortality are reduced, medical costs are reduced, and the hospitalization time is shortened [20].

## Analysis of the nutritional status of HAE patients and their relationship with clinical indicators

When BMI and ALB were used as evaluation indicators, the malnutrition rate of patients was 6.45% and 12.90% respectively, which was increased when NRS 2002 was used (36.56%). The sensitivity of BMI and ALB in assessing malnutrition of HAE patients is lower than that of NRS 2002. The disparity in the outcome from the different parameters may be because Tibetans in pastoral areas prefer high-protein beef, mutton and dairy products. Also, their body is better adapted to high altitude hypoxic environments, and their exercise intensity is higher. Considering their BMI and ALB, HGB baseline values are slightly higher than those of Han nationality people of the same age [21, 22]. Therefore, the NRS 2002 method is more reliable and prudent in assessing malnutrition of HAE patients, but it may also have a lower sensitivity. However, BMI and ALB can be used as screening indicators for clinical reference.

In this study, 36.56% of HAE patients were malnourished, which may be due to the following reasons:

1. The granulomatous reaction caused by *Echinococcus multilocular* can cause severe pathological damage to normal hepatocytes. Besides, HAE can cause various inflammatory cell infiltration and necrosis, while producing toxins that damage liver tissue, which causes extensive liver fibrosis [23]. Systematic infection and invasion of hepatocytes by *Echinococcus multilocular* cause disorders in nutrient synthesis and metabolism, resulting in malnutrition in HAE patients. The findings of this study revealed that a worse grading of liver function is associated with a lower ALB and a higher NRS2002 score. These findings indicate that the impairment of liver function caused by HAE is directly related to malnutrition in HAE patients.

2. In HAE patients, the continuous proliferation of liver lesions forms fibrous connective tissue "mass" and inflammatory granulation tissue, that depress the bile ducts and blood vessels in regions adjacent to the liver. These tissues protrude from the surface of the liver and squeeze the adjacent digestive tract, resulting in a series of complications, such as pain around the organ, portal hypertension, obstructive jaundice, bloating, and nausea. Vomiting and other discomforts [24] affects the patient's ability to eat and absorb nutrients from the digestive tract, further aggravating the poor nutritional status of patients. In this study, we found that the larger the transverse and longitudinal diameters of liver lesions in HAE patients, the higher the NRS2002 score ($P < 0.01$) and the lower the ALB value ($P < 0.01$). However, the NRS2002 score, BMI and ALB were not significantly correlated to the number and location of liver lesions. These results show that the larger the size of the injury, the more the invasion and compression of the adjacent bile ducts, blood vessels and digestive tract, which cause the complications mentioned above, and subsequent malnutrition.

3. Liver lesions in HAE patients stimulate the formation of adhesions within the nervous-rich liver capsule, which leads to chronic pain such as dull pain or swelling pain in the regions surrounding the liver. Subsequently, patients are more prone to depression, insomnia, anxiety and other negative emotions. Several other comprehensive psychological and physiological disorders, such as insomnia, oil-weariness, anorexia, long-term bed-rest, and reduced daily activities then arise. These disorders eventually lead to reduced diet regimes, slow gastrointestinal peristalsis, reduced nutrient absorption and utilization rates, and worsened the nutritional status of patients [25].

4. The inhabitants of the Tibetan plateau are highly prone to developing HAE. The pastoral Tibetans have backward economic conditions, poor hygiene quality, low education level and strong religious beliefs. Due to this combination of factors, the individuals are not keen to identify diseases at an early stage, resulting in an increase in chronic liver function damage by the time patients seek medical attention. This study found that there was no significant correlation between NRS2002 score, BMI, serum ALB value and blood AST, ALT, TBIL value ($P > 0.05$). However, AST, ALT and TBIL are indicators of acute liver damage, which indicates that the nutritional status of HAE patients is less related to whether they have acute liver damage but closely associated with chronic liver damage.

## Conclusions

Malnutrition is highly prevalent among hepatic alveolar echinococcosis (HAE) patients, and the nutritional status of HAE patients is related to many clinical factors, such as Child-Pugh

classification of liver function, lesion size, and serum albumin. While the body mass index (BMI) and albumin (ALB) can be used as primary screening indicators for malnutrition in HAE patients, the NRS 2002 method may be more reliable and prudent in assessing nutrition in HAE patients. Also, the NRS 2002 method is more suitable for clinical preoperative risk assessment and prediction of postoperative. The proper evaluation of the preoperative nutritional status of HAE patients is recommended in the concept of perioperative dietary support of Enhanced recovery after surgery (ERAS). Besides, the assessment minimizes the risks associated with anesthesia and surgery, shortens the hospitalization time, reduces the incidence of postoperative complications, and improves the prognosis of patients.

## Supporting information

**S1 Table. List of basic information of HAE patients.**
(XLS)

**S1 File. CONSORT 2010 flow diagram.**
(DOC)

**S2 File. PLOSOne_Clinical_Studies_Checklist.**
(DOCX)

**S3 File. STROBE_checklist_v4_combined_PlosMedicine.**
(DOCX)

**S4 File. Trendstatement_TREND_Checklist.**
(DOCX)

## Author Contributions

**Conceptualization:** Xie Liang.

**Data curation:** Wang Shu.

**Investigation:** Wang Shu, Gu Junqing.

**Project administration:** Li Jianshui.

**Resources:** Zhou Linyong, Dawa Enzhu.

**Writing – original draft:** Xie Liang.

**Writing – review & editing:** Xie Liang, Xu Mingqing.

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
