## [Decision Letter · Decision Letter 0]

23 Oct 2019

PONE-D-19-21935

Preoperative nutritional evaluation of patients with hepatic alveolar echinococcosis

PLOS ONE

Dear Mingqing,

Thank you for submitting your manuscript to PLOS ONE. After careful consideration, we feel that it has merit but does not fully meet PLOS ONE’s publication criteria as it currently stands. A major concern is that

your study did not compare the predictive effects of different nutritional assessment methods on surgical risk/clinical outcomes. Therefore, we invite you to submit a revised version of the manuscript that addresses the points raised during the review process. Please be informed that the revision does not guarrantee the acception for publication.

We would appreciate receiving your revised manuscript by Dec 07 2019 11:59PM. To enhance the reproducibility of your results, we recommend that if applicable you deposit your laboratory protocols in protocols.io, where a protocol can be assigned its own identifier (DOI) such that it can be cited independently in the future. For instructions see: http://journals.plos.org/plosone/s/submission-guidelines#loc-laboratory-protocols

We look forward to receiving your revised manuscript.

Kind regards,

Dong-Xin Wang

Academic Editor

PLOS ONE

**Journal Requirements:**

2. Please provide additional details regarding participant consent. In the ethics statement in the Methods and online submission information, please ensure that you have specified (1) whether consent was suitably informed and (2) what type you obtained (for instance, written or verbal). If your study included minors under age 18, state whether you obtained consent from parents or guardians. If the need for consent was waived by the ethics committee, please include this information.

3. We note that you have reported significance probabilities of 0 in places. Since p=0 is not strictly possible, please correct this to a more appropriate limit, eg 'p<0.0001'.

4. Thank you for sending us the data set underlying the results presented in your PLOS ONE submission. We notice that some of the information included in the data set may be potentially identifying. Please ensure that the data shared are in accordance with participant consent and provide only the data that are used in this specific study. To ensure patient confidentiality, we would recommend removing the columns with patient names. Additional guidance on preparing raw clinical data for publication can be found in our Data Policy FAQs (https://journals.plos.org/plosone/s/data-availability#loc-clinical-data).

5. Thank you for stating that “The funders had no role in study design, data collection and analysis, decision to publish, or preparation of the manuscript” in your financial disclosure.

Please also provide the name of the funders of this study (as well as grant numbers if available) in your financial disclosure statement.

**Comments to the Author**

1. Is the manuscript technically sound, and do the data support the conclusions?

Reviewer #1: Partly

Reviewer #2: Yes

2. Has the statistical analysis been performed appropriately and rigorously? 

Reviewer #1: Yes

Reviewer #2: Yes

3. Have the authors made all data underlying the findings in their manuscript fully available?

Reviewer #1: Yes

Reviewer #2: Yes

4. Is the manuscript presented in an intelligible fashion and written in standard English?

Reviewer #1: Yes

Reviewer #2: No

5. Review Comments to the Author

Reviewer #1: In the current manuscript, the authors evalued the nutritional status of 93 HAE patients retrospectively, to find a concise and reasonable nutritional evaluation indicator to serve the clinical preoperative risk assessment. And the results of the study showed that NRS 2002 may be more reliable and prudent than that of BMI or ALB level in judging malnutrition in HAE patients, and the nutritional status of HAE patients is related to many clinical factors, such as Child-Pugh classification of liver function, size of lesion, ALB and so on. The authors concluded that NRS 2002 is more suitable for preoperative risk assessment for HAE patients.

Overall, the current manuscript showed us some impressive information, e.g. "malnutrition is common in HAE patients", "compared to BMI and ALB, NRS 2002 can help diagnose more malnutrition in HAE patients".

However, there are still some issues that must be clarified to support the current conclusion drawn from the available results:

1. The purpose of the current study was to "find a concise and reasonable nutritional evaluation indicator" to find out the HAE patients with malnutrition, which could help serve the clinical preoperative risk assessment better, to guide preoperative nutritional support treatment and to reduce the risk of surgery for treatment of HAE. However, in the current study they did not compare the predictive effects of different nutritional assessment methods on surgical risk. It seems that they just considered NRS 2002 as the gold standard for diagnosis of malnutrition.

2. As mentioned before, no comparison was made to evaluate the predictive effects of different nutritional assessment methods (e.g. BMI, ALB, NRS 2002) on surgical risk of HAE, why the authors concluded that "NRS 2002 is more suitable for preoperative risk assessment for HAE patients"? Although NRS 2002 could diagnose more malnutrition patients than the other methods, this also means that with this indicator more HAE patients were not suitable for direct surgical treatment, unless a preoperative nutritional support was introduced. Is this consistent with the current status of HAE treatment (about 36.56% of the HAE patients should receive preoperative nutritional support before operation)? If so, the authors should state that point to emphasize the significance of this study, otherwise the results should be reinterpreted carefully, that maybe ALB or BMI, but not NRS 2002, is more suitable for preoperative risk assessment for HAE patients.

Reviewer #2: Xie et al. in their manuscript entitled "Preoperative nutritional evaluation of patients with hepatic alveolar echinococcosis" reported preoperative nutritional evaluation for patients with hepatic alveolar echinococcosis (HAE). This is a large sample-size research, as the HAE is rare disease and malnutrition did not get much attention for the investigators. The authors found malnutrition is common event, and several evaluation methods had subtle difference. While, several minor concerns may need to be addressed:

1. It is important to analysis the hemoglobin, wound complications and hospital stay for malnutrition patients.

2 As the present study enrolled 93 patients, it would be interesting if the author stratify those patients into the poor nutrition and good nutrition groups.

3. The conclusion should be more concise to show the goal of this study.

4. The abbreviations should be spelt out in full name the first time. This manuscript needs to be polished by an English-native speaker.

6. PLOS authors have the option to publish the peer review history of their article (what does this mean?). If published, this will include your full peer review and any attached files.

Reviewer #1: Yes: Yinmo Yang

Reviewer #2: No

---

## [Author Response · Author response to Decision Letter 0]

14 Nov 2019

Response to Reviewers

Dear DongXin and reviewers,

We thank you and the reviewers for reviewing our manuscript entitled "Preoperative nutritional evaluation of patients with hepatic alveolar echinococcosis" (ID：PONE-D-19-21935). The reviewers comments have considerably helped to revise and improve our paper, in addition to providing important guidance to our research. We have studied the comments carefully and made corrections which we hope meet your approval. The main corrections are incorporated in the manuscript and responses to the reviewers’ comments are provided below.

Replies to the Journal Requirements

Response：Several changes and adjustments have been made to comply with PLOS ONE's style requirements, including those for file naming. Thank you.

2.Please provide additional details regarding participant consent. In the ethics statement in the Methods and online submission information, please ensure that you have specified (1) whether consent was suitably informed and (2) what type you obtained (for instance, written or verbal). If your study included minors under age 18, state whether you obtained consent from parents or guardians. If the need for consent was waived by the ethics committee, please include this information.

Response：Thank you for pointing this out. Additional details concerning participants consent have been added in the ethics statement and the corresponding content has been added to the new 'cover letter'.

3. We note that you have reported significance probabilities of 0 in places. Since p=0 is not strictly possible, please correct this to a more appropriate limit, eg 'p<0.0001'.

Response：Thank you for the suggestion. This has been changed to 'p<0.001' in the manuscript. 

4. Thank you for sending us the data set underlying the results presented in your PLOS ONE submission. We notice that some of the information included in the data set may be potentially identifying. Please ensure that the data shared are in accordance with participant consent and provide only the data that are used in this specific study. To ensure patient confidentiality, we would recommend removing the columns with patient names. Additional guidance on preparing raw clinical data for publication can be found in our Data Policy FAQs (https://journals.plos.org/plosone/s/data-availability#loc-clinical-data).

Response：I've deleted unnecessary columns in the data to ensure patient confidentiality, and named it 'S1 Table', and uploaded it again as Supporting information.

5. Thank you for stating that “The funders had no role in study design, data collection and analysis, decision to publish, or preparation of the manuscript” in your financial disclosure.

Please also provide the name of the funders of this study (as well as grant numbers if available) in your financial disclosure statement.

Response：We have added a financial disclosure statement to the new 'cover letter'.

We also bring to your attention that, after careful discussion, we have revised the signature unit of the first author, because Xie Liang participated in this research at a time when support from the Affiliated Hospital of North Sichuan Medical College to Ganzi County People's Hospital(Tibet Aid Project) was provided, and the supporting unit of fund source is also the Affiliated Hospital of North Sichuan Medical college, so the Affiliated Hospital of North Sichuan Medical College should be the first completion unit of this paper. We apologize for the inconvenience. 

6.Please include captions for your Supporting Information files at the end of your manuscript, and update any in-text citations to match accordingly. Please see our Supporting Information guidelines for more information: http://journals.plos.org/plosone/s/supporting-information.

Response： Thank for this suggestion. We have added Supporting Information files at the end of my manuscript, renamed and uploaded the corresponding files. 

Replies to the Reviewer #1:

1.The purpose of the current study was to "find a concise and reasonable nutritional evaluation indicator" to find out the HAE patients with malnutrition, which could help serve the clinical preoperative risk assessment better, to guide preoperative nutritional support treatment and to reduce the risk of surgery for treatment of HAE. However, in the current study they did not compare the predictive effects of different nutritional assessment methods on surgical risk. It seems that they just considered NRS 2002 as the gold standard for diagnosis of malnutrition.

Response： Thank you for the advice. We carried out a sub-comparison study to assess the predictive effects of different nutritional assessment methods on surgical risk. Please see the revised manuscript for details.

Specifically, we also noted that according to BMI or ALB group, the hospitalization time of malnutrition group was slightly longer than that of normal nutrition group, and the HGB on the 3rd day after operation was slightly lower than that of normal nutrition group, but both these differences were not statistical significant. For this regard, we think that, on one hand, the sample size may be insufficient, but at the same time, BMI and ALB are less sensitive to the prediction of postoperative recovery. 

2. As mentioned before, no comparison was made to evaluate the predictive effects of different nutritional assessment methods (e.g. BMI, ALB, NRS 2002) on surgical risk of HAE, why the authors concluded that "NRS 2002 is more suitable for preoperative risk assessment for HAE patients"? Although NRS 2002 could diagnose more malnutrition patients than the other methods, this also means that with this indicator more HAE patients were not suitable for direct surgical treatment, unless a preoperative nutritional support was introduced. Is this consistent with the current status of HAE treatment (about 36.56% of the HAE patients should receive preoperative nutritional support before operation)? If so, the authors should state that point to emphasize the significance of this study, otherwise the results should be reinterpreted carefully, that maybe ALB or BMI, but not NRS 2002, is more suitable for preoperative risk assessment for HAE patients.

Response：Thank you for the advice. We have made new comparisons for postoperative recovery indicators between normal nutrition group and malnutrition group which has been sub-grouped by NRS2002. Please see the new manuscript for details.

According to NRS 2002 nutrition score, preoperation HGB was not different between the two groups (P＞0.05). The hospitalization time of normal nutrition group was significantly shorter than that of malnutrition group (P < 0.05), HGB on the 3rd day after operation was significantly higher, and the number of postoperative complications was lower than in the malnutrition group (P < 0.05). 

Replies to the Reviewer #2:

1. It is important to analysis the hemoglobin, wound complications and hospital stay for malnutrition patients.

Response：Thank you for the advice. The hospitalization time, HGB value on the 3rd day after operation and the number of postoperative complications (including wound liquefaction and infection, lung infection, abdominal and liver wound infection, urinary tract infection, postoperative inflammatory intestinal obstruction, bile leakage and acute liver function injury) were recorded and analyzed in the new manuscript. Please see the revised manuscript for details.

2. As the present study enrolled 93 patients, it would be interesting if the author stratify those patients into the poor nutrition and good nutrition groups.

Response：Thank you for the advice. All patients have been divided into the normal nutrition group and the malnutrition group according to NRS 2002, BMI and ALB. Please see Table 6 for the comparative analysis of postoperative recovery indicators between the normal nutrition group and the malnutrition group.

According to NRS 2002 nutrition score, preoperation HGB did not differ between the two groups (P＞0.05). The hospitalization time of normal nutrition group was significantly shorter than that of malnutrition group (P < 0.05). HGB on the 3rd day after operation was significantly higher than in the malnutrition group (P < 0.05), and the number of postoperative complications was lower than in the malnutrition group (P < 0.05). 

3. The conclusion should be more concise to show the goal of this study.

Response：Thank you for the suggestion. We have revised the conclusion of the manuscript to make it more concise.

4.The abbreviations should be spelt out in full name the first time. This manuscript needs to be polished by an English-native speaker.

Response：Thank you for pointing this out. The abbreviations have been spelt out in full name at the first use. We have sought the services of a professional editing company to improve the readability of the paper.

Once again, we appreciate your constructive comments and suggestions which have helped to improve the clarity and depth of the paper.

Yours sincerely,

Xie Liang 

Email address: scuxl@foxmail.com

Corresponding author: Xu Mingqing

Email address: xumingqing@scu.edu.cn

Email address: scuxl@foxmail.com

Corresponding author : Xu Mingqing

Email address: xumingqing@scu.edu.cn

---

## [Decision Letter · Decision Letter 1]

6 Feb 2020

Preoperative nutritional evaluation of patients with hepatic alveolar echinococcosis

PONE-D-19-21935R1

Dear Dr. Xu Mingqing,

We are pleased to inform you that your manuscript has been judged scientifically suitable for publication and will be formally accepted for publication once it complies with all outstanding technical requirements.

With kind regards,

Dong-Xin Wang

Academic Editor

PLOS ONE

Additional Editor Comments (optional):

Reviewers' comments:

Reviewer's Responses to Questions

**Comments to the Author**

1. If the authors have adequately addressed your comments raised in a previous round of review and you feel that this manuscript is now acceptable for publication, you may indicate that here to bypass the “Comments to the Author” section, enter your conflict of interest statement in the “Confidential to Editor” section, and submit your "Accept" recommendation.

Reviewer #1: All comments have been addressed

Reviewer #2: All comments have been addressed

2. Is the manuscript technically sound, and do the data support the conclusions?

Reviewer #1: Yes

Reviewer #2: Yes

3. Has the statistical analysis been performed appropriately and rigorously? 

Reviewer #1: Yes

Reviewer #2: Yes

4. Have the authors made all data underlying the findings in their manuscript fully available?

Reviewer #1: Yes

Reviewer #2: Yes

5. Is the manuscript presented in an intelligible fashion and written in standard English?

Reviewer #1: Yes

Reviewer #2: Yes

6. Review Comments to the Author

Reviewer #1: All my comments have been fully addressed, and the current manuscript looks to meet the publication requirements.

Reviewer #2: The authors have satisfactorily addressed the comments and the manuscript has been improved in the revised manuscript.

7. PLOS authors have the option to publish the peer review history of their article (what does this mean?). If published, this will include your full peer review and any attached files.

Reviewer #1: No

Reviewer #2: No

---

## [Editor Report · Acceptance letter]

10 Feb 2020

PONE-D-19-21935R1 

Preoperative nutritional evaluation of patients with hepatic alveolar echinococcosis 

Dear Dr. Mingqing:

I am pleased to inform you that your manuscript has been deemed suitable for publication in PLOS ONE. Congratulations! Your manuscript is now with our production department. 

With kind regards,

on behalf of

Professor Dong-Xin Wang 

Academic Editor

PLOS ONE